# Impact of a UV-C Scalable Reactor on the Chemical and Sensory Quality of Peppercorns

**DOI:** 10.3390/foods14173056

**Published:** 2025-08-29

**Authors:** Víctor Arcos-Limiñana, Soledad Prats-Moya, Salvador Maestre-Pérez

**Affiliations:** Department of Analytical Chemistry, Nutrition and Food Sciences, University of Alicante, 03690 Alicante, Spain; victor.arcos@ua.es (V.A.-L.); maria.prats@ua.es (S.P.-M.)

**Keywords:** UV-irradiation, sensory analysis, non-thermal processing, food quality, black pepper, spices, nutritional composition

## Abstract

This study investigates the chemical and sensory effects of UV-C disinfection on black, white, green, and pink peppercorns using a scalable mechanical drum reactor. While previous research has demonstrated the efficacy of UV-C radiation in microbial disinfection, there is a lack of deep, quality-focused research on food products. Nevertheless, for spices, this is just as important, if not more so, than food safety. Different analyses were conducted to assess changes in volatile compounds, organic acids, fatty acids, tocopherols, and colour following UV-C exposure. Additionally, sensory evaluations were performed using triangular tests to determine whether these chemical changes were perceptible to consumers. Results revealed that many of the measured chemical components were affected by the UV treatment, with some volatile compounds decreasing by up to 90%, while certain organic acids increased by more than 150%. Despite these changes, no significant differences in colour, aroma, or flavour were detected by the sensory panel across all pepper types. These findings suggest that UV-C irradiation, when applied under the tested conditions, preserves the sensory quality of peppercorns, supporting its potential as a non-thermal processing method for spice treatment.

## 1. Introduction

The application of ultraviolet (UV) radiation for the disinfection of spices has been a popular subject of research in recent years, with studies conducted on a range of spices, including oregano [1], paprika [2], and black pepper [3], among others. Given UV light’s low penetration power, only the surface directly exposed to the radiation is effectively disinfected, necessitating some sample movement to ensure full surface exposure. Two main types of reactors have been employed for this purpose: mechanical reactors and fluidised beds. Although fluidised beds have shown promising results [1], their complex implementation, limited load capacity, and high installation and maintenance costs have hindered widespread adoption [2]. In contrast, mechanical reactors have been extensively used across various industrial applications [3] and can be easily integrated into existing production lines.

Black pepper is one of the most consumed spices globally [4], and it is also the most frequently reported spice in the Rapid Alert System for Food and Feed (RASFF) of the European Union due to microbiological contamination [5]. The primary gastronomic properties of peppers are their smell and flavour, which thermal disinfection methods can compromise; therefore, an alternative method would be beneficial. Black peppercorns are an ideal candidate for UV disinfection due to their specific characteristics. Peppercorns are robust and can withstand the movement that UV treatments require. They have a high surface-to-mass ratio and a nearly rounded shape, allowing minimal space for bacteria to hide from UV radiation. Dry white, green, and pink peppercorns exhibit comparable characteristics and challenges. However, despite their suitability for UV disinfection, no published studies have yet explored this potential.

Previous studies have indicated that both UV light and reactor design may induce chemical changes [6,7,8], potentially leading to sensory alterations. However, these studies often rely on preliminary indicators such as total phenolic content and antioxidant activity, which, while helpful in identifying trends, do not provide a comprehensive understanding of the underlying changes. Sensory analysis is critical in both industry and consumer perception, as it directly influences purchasing decisions. Nevertheless, no sensory evaluation has been conducted on UV-treated peppercorns to date.

Therefore, the objective of this study was to perform a comprehensive analysis of the chemical changes in peppercorns following UV-C disinfection using a mechanical drum reactor. This study involved detailed assessments of volatile compounds, organic acids, tocopherol profiles, and fatty acids, along with colourimetry, across black, green, white, and pink peppercorns. Additionally, triangular sensory tests were conducted to evaluate perceptible changes in colour, aroma, and flavour of the peppercorns resulting from the UV treatment.

## 2. Materials and Methods

### 2.1. Reactants

Cis-3-hexenyl acetate (>98%, Thermo scientific, Kandet, Germany), vanillin (99%, Sigma Aldrich, St. Quentin Fallavier, France), succinic acid (>99.5%, Sigma Aldrich, Wien, Austria), caffeic acid (>98%), quercetin (>95%) and trans-ferulic acid (>99%) (Sigma Aldrich, Shanghai, China), ethanol (96%) (VWR, Leuven, Belgium), (-)-epicatechin (>99%), (+)-catechin (>99%), D-(-)-quinic acid (98%), p-hydroxybenzoic acid (>99%), protocatechuic acid (>99%), p-coumaric acid (>98%), trans-cinnamic acid (>99%), vanillic acid (>97%) and syringic acid (98%) (Sigma Aldrich, Taufkirchen, Germany), n-hexane (>99%, Sigma Aldrich, St. Louis, MO, USA), sodium methoxide (25% *w*/*v*, Sigma Aldrich, St. Louis, MO, USA), sulphuric acid (96%, Panreac, Barcelona, Spain), FAME 37 multistandard, alpha-tocopherol (neat), gamma-tocopherol and delta-tocopherol (neat) (Supelco, Bellefonte, PA, USA).

### 2.2. Samples

Four types of dried peppercorns from different geographical origins were analysed: black and white pepper from Vietnam, green pepper from India, and pink pepper from Brazil, collected in 2023, provided by a Spanish spices distributor (Jesús Navarro S.A.). Notably, black, white, and green peppers are derived from the *Piper nigrum* plant, while pink pepper is obtained from *Schinus terebinthifolius*.

### 2.3. UV-C Treatment

The setup consisted of a low-pressure mercury vapour lamp producing 2.5 W in the UV-C spectrum (TUV PL-S 9W, Philips, Amsterdam, The Netherlands), with a wavelength peak of 254 nm, coupled to a reactor and located 5 cm above the sample. The reactor consisted of a hexagonal rotary plastic drum with an internal diameter of 95 to 105 mm, a length of 140 mm, and a rotation speed of 20 rpm, with a change in the rotation direction occurring every 30 s (Figure 1). This type of reactor has previously proven successful in achieving a consistent UV distribution on the surface of peanuts [9]. The treatment duration was set to 90 min, based on the findings of a preceding experiment that explored the effective microbial inactivation [10]. In each instance, 15 g of a single type of pepper was irradiated.

A UV-C PCE-UV36N radiometer sensor (PCE Instruments, Tobarra, Spain) for mercury lamps (220–280 nm) was placed at the same height as the samples inside the reactor to replicate the experimental conditions. Before measurements, the UV lamp was left on for 30 min to stabilise the irradiance, and the average irradiance was recorded over this period. The average irradiance was found to be 5.51 ± 0.09 mW/cm^2^, which equates to 29.8 ± 0.5 J/cm^2^ for the 90 min treatment. A direct temperature measurement was conducted on the sample after 90 min of treatment, resulting in a rise of only 5 °C. Three individual treatments were performed for each sample and then analysed.

### 2.4. Colour Measurement

The colour of the samples was assessed using a Chroma Meter CR-400 (Konica Minolta, Tokyo, Japan). The samples, both treated and untreated, were ground using a coffee grinder for 45 s or until the powder was homogenous and then placed into the accessory CR-A50, which is designed explicitly for granular and powdered materials. The accessory consists of a small black chamber with a glass that fits into the colourimeter. The colourimeter was calibrated utilising the blank standard plate supplied by the manufacturer (L*: 97.63, a*: −0.48 and b*: 2.12). Six replicates of every sample were assessed to obtain the CIELAB values—L* (luminosity), a* (redness) and b* (yellowness)—and to calculate the total colour difference (∆E) among samples using the following equation:

(1)$$∆E\;=\;\surd \;({\left({L^{*}}_{2}\;-\;{L^{*}}_{1}\right)}^{2}\;+\;{\left({a^{*}}_{2}\;-\;{a^{*}}_{1}\right)}^{2}\;+\;{\left({b^{*}}_{2}\;-\;{b^{*}}_{1}\right)}^{2})$$
where L*_1_, a*_1_ and b*_1_ are the parameters obtained from the control sample, and L*_2_ a*_2_ and b*_2_ are the parameters obtained from the treated sample.

### 2.5. Volatile Compounds Determination

Following a modified version of the methodology described by Baky et al. [11], 100 mg of sample was placed in 15 mL SPME screw-cap vials, spiked with 5 µL of cis-3-hexenyl acetate as an internal standard (IS). The volatile analysis method developed by Juan-Polo et al. was adapted for this application [12]. The separation of volatile organic compounds was performed on an Agilent 7890B gas chromatograph (Palo Alto, CA, USA) coupled with an Agilent 5977B mass spectrometer (Palo Alto, CA, USA). Compounds were separated on a fused silica capillary J&W DB-624 GC Column (30 m, 0.25 mm i.d., 1.40 µm film thickness), purchased from Agilent (Palo Alto, CA, USA). Helium was used as carrier gas (1 mL/min). The injector port and interface temperatures were 250 °C. Samples were heated at 75 °C for 25 min. Afterwards, the SPME fibre (DVB/CAR/PDMS SPME, 50/30 m, StableFlex, 1 cm long) fixed on the mechanical arm was exposed to the headspace of the sample vial for extraction of the compounds. After 60 min of exposure, the fibre was withdrawn and introduced into the GC injector for desorption (3 min) and analysis. The column temperature was programmed to be 70 °C (held for 2 min) and then increased to 250 °C (held for 12 min) at a rate of 5 °C/min. Samples were injected into the column with a 1:50 split. Acquisition parameters of the Mass Spectrometer were: electron energy, 70 eV; source temperature, 230 °C; quadrupole temperature, 150 °C; mass range *m*/*z* 15–350; scan rate, 3.62 s/scan; and EM voltage, 1190 V. Identification of compounds was performed comparing the mass spectra obtained from the GC/MS analysis with the spectra found in the National Institute of Standards and Technology (NIST) database.

To evaluate the contribution of the movement of the sample to the modification of the volatile profile, an agitated control experiment was also conducted. In this experiment, the samples were subjected to the reactor’s agitation as the treated samples, but the UV lamp was not activated, observing the composition of agitated-only samples and comparing them to control and UV-treated peppers.

Standardisation of the signal was obtained by dividing the raw area of each peak by the IS area and then by the mass of the sample in grams. All volatile data analyses were performed using the standardised signal per gram of sample, and were expressed as retention percentage in the graphs, which was calculated by dividing the standardised signal of the compound in the UV-treated pepper by that in a control sample.

### 2.6. Flavonoids, Polyphenols and Other Organic Acids Determination

For the analysis of the organic acid composition, an ethanolic extraction was required. One hundred milligrams of ground pepper was vortexed with 5 mL of 70% ethanol for one minute. Subsequently, the mixture was centrifuged at 4200× *g* for 10 min, and the supernatant was collected. The pellet was resuspended in 5 mL of 70% ethanol, and the extraction was repeated, mixing both supernatant extracts. All extractions were conducted in triplicate. The extracts were filtered through a 0.45 µm pore size membrane (Filtros Anoia, Barcelona, Spain) and stored at −15 °C in the freezer until analysed.

The organic acid profile analysis was performed using an Agilent 1290 Infinity Online UHPLC with a triple quadrupole spectrometer featuring JetStream and iFunnel technology (UHPLC-1290/QQQ-6490; Agilent, Palo Alto, CA, USA). Five microliters of the filtered extract were injected into the UHPLC and eluted at a flow rate of 0.5 mL/min using a Poroshell 120 EC-18 2.7 μm, 3 × 100 mm column (Agilent, Palo Alto, CA, USA). The gradient and solvent composition are shown in Table 1. Negative and positive ion modes were utilised using the following parameters: gas temperature of 275 °C and a nebuliser pressure of 40 psi. Standard curves were obtained employing different dilutions of stock solutions containing from 8 to 0.01 ppm of every analysed compound (epicatechin, catechin, vanillin, succinic acid, trans-ferulic acid, p-hydroxybenzoic acid, caffeic acid, quercetin, protocatechuic acid, gallic acid, trans-cinnamic acid, p-coumaric acid, syringic acid, vanillic acid and quinic acid). Additionally, neochlorogenic acid, chlorogenic acid, rutin, phlorizin, kaempferol and hesperidin were qualitatively analysed. Results were expressed as retention percentage in the graphs, which was calculated by dividing the concentration of the compound in the UV-treated pepper divided by that in a control sample.

### 2.7. Fatty Acid and Tocopherol Determination

The fatty acid and tocopherol composition was determined using an adapted version of the direct analysis methodology of Juan-Polo et al. [13]. Briefly, 3 mL of n-hexane containing 150 ppm BHT was added to 1 g of ground pepper, which was then mixed with 50 mg of ascorbic acid to prevent tocopherol oxidation. This mixture was then stirred with a magnetic stirrer at 900 rpm and 50 °C for 10 min. This procedure was repeated after adding 1.5 mL of 0.2 M sodium methoxide, and one more minute after adding 1.5 mL of a 5% *v*/*v* sulfuric acid solution in methanol. Lastly, the supernatant was collected and filtered using a 0.22 μm nylon-membrane syringe filter. The filtered supernatant was diluted 1:10 with hexane for the fatty acid methyl esters (FAMEs) analysis, but no dilution was employed for the tocopherol analysis.

FAMEs were determined using an Agilent 7890N GC coupled to a 5977B Mass Spectrometer (MS) (Agilent Technologies, Palo Alto, CA, USA). The column used was a BPX70 column (60 m × 0.25 mm × 0.25 μm). Acquisition parameters: Inj man. = 270 °C, Volume = 1.00 μL, Split = 10:1, Carrier Gas = He, Solvent Delay = 7.80 min, Transfer Temp = 250 °C, Source Temp = 250 °C, Scan: 35 to 350 Da. Helium flow rate was fixed at 1 mL/min, and samples were injected using a 1:100 split ratio. FAMEs were identified using the National Institute of Standards and Technology (NIST) MS library matches. FAME standards consisted of a series of dilutions ranging from 2 to 150 ppm of each of the 37 FAME mixtures.

Tocopherol determination was performed using an Agilent 1260 Infinity Binary System ultra-high-performance liquid chromatography (UHPLC) system with a fluorescence detector (Agilent Technologies, Palo Alto, CA, USA). Separation of target analytes was performed using an Inertsil NH2 NP-HPLC column (5 μm, 250 × 4.6 mm I.D.) thermostated at 40 °C. A mixture of n-hexane: isopropanol (98:2 *v*/*v*) was used as mobile phase at a flow rate of 1.2 mL/min in isocratic mode, with the sample injection volume of 20 μL. Fluorescence absorption and emission wavelengths were fixed at 298 nm and 325 nm, respectively. Tocopherols in samples were identified by comparison to the retention times of the standards. The range of the calibration standard was between 0.1 and 15 ppm for each tocopherol.

### 2.8. Sensory Analysis

The analysis was conducted in accordance with the ISO 4120:2021 standard [14] for triangle tests in sensory evaluation. Triangle tests were employed to identify perceptible differences between treated and untreated pepper samples across three sensory attributes: colour, smell, and flavour. Each triangle test consisted of a set of three subsamples, each labelled with a randomly generated three-digit code. Among the three subsamples, one differed from the other two—one treated and two untreated samples or vice versa—and participants were asked to identify the distinct pepper. Each set of triangle tests included evaluations of colour, smell, and flavour, and was conducted using only one type of pepper (black, green, white, or pink) at a time.

For the colour assessment, three vials of 13.5 mL were used, each containing 1 g of peppercorns. A separate set of three vials, also containing 1 g of peppercorns each, was prepared for the smell evaluation. For the flavour assessment, three servings of long white rice were used, as described by Dawodu et al. [15], each seasoned with either treated or untreated finely ground pepper. The rice servings were maintained at 45 °C in an oven RATIONAL Icombiclassic (Landsberg am Lech, Germany) and served immediately before the flavour evaluation. Each rice serving contained between 8 and 10 g of rice, with a seasoning ratio of 0.1 g of pepper and 1.5 g of salt per 100 g of rice. Participants were provided with a cup of water and instructed to rinse their mouths after tasting each rice sample to avoid cross-sample interference. In addition to identifying the distinct sample in each triangle test, participants were asked to provide a basic qualitative evaluation of the smell and flavour of the samples in comparison to the other two. The order in which the three samples were presented to each participant was randomised to minimise order bias. All tests were conducted in the morning, between 10:00 and 13:00. To ensure unbiased data analysis, participant identities were anonymized.

A convenience sample of 58 participants was recruited from the University of Alicante, Spain, either in person or via email. The sample included 37 women and 21 men, ranging in age from 18 to 65 years, with a mean age of 28.4 years, all of whom were untrained panellists. Prior to participation, each individual received a printed informed consent form and was free to withdraw from the study at any time. Participants with any medical condition—whether temporary or permanent—that could interfere with the accurate evaluation of sensory attributes were excluded before the testing began.

All participants were required to complete all tests on the same day. To be included in the final analysis, each participant had to respond to all three sensory attributes (colour, smell, and flavour) for at least two pepper types. A total of 40 valid responses per sample were obtained, in accordance with the ISO 4120:2021 standard parameters (α = 0.05; β = 0.2; pd = 30%). Each pepper type was tested using ten sets of subsamples, half of which contained two untreated and one treated sample.

The sensory analysis protocol was reviewed and approved by the Ethical Committee of the University of Alicante (Ref. 2023-12-21).

## 3. Results and Discussion

### 3.1. Colour

After UV exposition, significant colour differences were observed in all peppercorns, as shown in Table 2. The highest colour difference was observed in pink pepper after 90 min of treatment, with an ∆E of 4.4, while in *P. nigrum* peppercorns, a value of no more than 1.9 was observed. The primary contributor to the ∆E was a* for *P. nigrum* peppercorns, but for pink pepper, all three parameters contributed to increasing the colour difference.

The data do not reveal a consistent trend in total colour difference across pepper types, indicating that intrinsic variability in food matrices may exert a greater influence on colour variation than the UV-C treatment itself. This observation aligns with previous studies, which have reported minimal or negligible colour changes following UV-C irradiation [8,16,17,18,19,20]. For instance, one study reported no significant colour alteration after up to 30 min of exposure [8], and even at elevated doses (76 J/cm^2^), only minor changes were detected [16]. Conversely, an ∆E increase of 4.5 was documented in black pepper subjected to a 31 J/cm^2^ pulsed UV treatment, although this was attributed to surface burning caused by the pulsed light [6]. In contrast, another investigation involving pulsed UV treatment reported no measurable impact on colour [21], further underscoring the variability in outcomes depending on the irradiation method and matrix characteristics.

### 3.2. Volatile Composition

The qualitative analysis of the volatile compounds revealed different patterns of compositional change among pepper types after 90 min of UV-C treatment. From the total detected peaks, nineteen compounds were selected based on signal intensity and smell relevance as the inclusion criteria [7,11,22,23,24].

Minor alterations were noted in the volatile profile of black pepper, as illustrated in Figure 2. Although statistical significance was observed in thirteen of the identified compounds, only α-phellandrene and α-terpinolene exhibited reductions exceeding 15%, and these compounds are considered minor constituents in black pepper. A comparable pattern was observed in green peppercorns, with the majority of volatiles remaining close to their original concentrations. In contrast, white pepper demonstrated greater susceptibility to UV-C treatment, with eight compounds showing significant reductions ranging from over 15% to as much as 65% of their initial concentrations. Pink pepper exhibited the most pronounced changes, with several volatile compounds nearly completely degraded.

Among the three *P. nigrum* peppercorns, green pepper was characterised by elevated levels of sabinene, β-pinene, and α-pinene, whereas black pepper contained minimal 3-carene but high concentrations of α-copaene, β-bisabolene, and caryophyllene. White pepper exhibited higher levels of 3-carene and α-phellandrene. Notably, black pepper was the only sample in which germacrene-D was detected, and it also contained the highest levels of α-copaene and caryophyllene. Pink pepper displayed a markedly different volatile profile, with substantially higher concentrations of 3-carene, D-limonene, myrcene, α-phellandrene, and α-terpinolene. The standardised area per gram of volatile compounds for each sample is provided in Appendix A.

White pepper was more susceptible to the treatment’s effects than green and black pepper. The increased sensitivity of white pepper to UV-C treatment may be attributed to the absence of its pericarp, which typically serves as a protective barrier. Removal of the pericarp exposes the endocarp, facilitating the evaporation of volatile compounds. Although pink pepper does present pericarp, it is easily broken, and a similar phenomenon could have occurred to it.

While no direct reaction analysis was conducted in this study, the degradation of volatile compounds is likely due to mechanisms such as UV-induced cleavage of double bonds, the generation and activity of reactive oxygen species (ROS), photoisomerization, and matrix-specific interactions [2]. The observed increase in cadiene may result from enzymatic activation of precursor compounds, a phenomenon known as hormesis, which has been documented in various matrices following UV exposure, including post-harvest treatments [2].

To isolate the effect of mechanical agitation from UV-C exposure, a control experiment was conducted in which samples were agitated under identical conditions without UV irradiation. This experiment revealed negligible changes in the volatile profiles of most pepper types. In black pepper, only α-terpinolene showed a significant reduction of 36%, while in green pepper, α-phellandrene decreased by 34% due to agitation alone, and was further reduced by UV-C treatment. No other significant changes were observed in the volatile profiles of black, green, white, or pink peppercorns, indicating that the majority of alterations were attributable to UV-C exposure rather than mechanical agitation. The results of this control experiment are also presented in Appendix A.

Previous studies have also reported volatile loss after UV treatments. Similarly to what is observed in the present work, a UV treatment resulted in a decrease in many volatiles of oregano, mainly caryophyllene and β-bisabolene [25], although these were generally unaffected in peppercorns. In one study, black peppercorns were irradiated with different wavelengths and doses [7]. An increase in caryophyllene was observed after the maximum treatment time of 20 min, using 280 and 365 nm irradiation, indicating that the rise in caryophyllene was radiation-induced. However, no increase was observed in our results. Decreases in 3-carene, limonene, β-pinene, and α-phellandrene were observed in UV LED-treated samples, which partially corresponds to our results, supporting the theory that major sesquiterpenes are light-responsive [7]. In grapes [26] and fruit juices [27,28], UVC was observed to retain or even increase the amount of linalool, which provides a floral aroma, but no increase was observed in UV-treated peppercorns.

### 3.3. Flavonoids, Phenolic Acids and Other Organic Acids

The quantification of different compounds of these groups in peppercorns is shown in Appendix A, and the qualitative analysis of 6 additional components is shown in Appendix A. Regarding the raw samples, it is noteworthy that in *P. nigrum* peppers, the major quantified component was succinic acid, followed closely by cinnamic acid in white pepper. In contrast, gallic acid was the main organic acid in pink peppercorns.

On the other hand, most of the compounds were not significantly impacted by the UV treatment, although some significant changes were observed, as shown in Figure 3. Gallic acid showed the most significant increase, nearly tripling its concentration after irradiation in black pepper. A considerable increase was also observed in white pepper, although the treatment caused the gallic acid to drop below 20% of its original content in green pepper. Quinic acid concentrations increased by approximately 38% after UV treatment in both white and green peppers. Additionally, protocatechuic acid and vanillic acid were significantly increased in white pepper, whereas in green pepper, it was neochlorogenic acid. Although neochlorogenic acid almost doubled in green pepper after the treatment, it could not be quantified.

Caffeic acid, on the other hand, was observed to decrease after treatment to 25% in black pepper but increase to 70% in pink pepper. Despite these changes, it is worth noting that gallic, caffeic, neochlorogenic, and quinic acids were minor compounds and may have had a limited impact on sensory quality. No pattern in organic acid changes could be observed between peppers, as they exhibited different behaviours following UV exposure. The changes in the remaining compounds were not significantly different after the UV treatment.

As with volatile compounds, the degradation of the compounds is likely to have happened due to the cleavage of double bonds hit by UV photons, the formation and action of ROS, photoisomerization and a range of matrix effects [2]. The increase in some of the phenolic compounds in peppers has been observed previously [10], and it has been related to the activation of enzymes converting precursors but can also be due to the breakage of polyphenols into smaller molecules [2].

Although several articles have reported changes in total phenolic content in similar matrices [4,28,29], this is the first work to assess the changes in specific phenolic compounds after a 254 nm UV treatment, to the best of the authors’ knowledge. Hernandez-Aguilar et al. [30] irradiated turmeric bread and observed that an increase in chlorogenic, ferulic, protocatechuic, p-hydroxybenzoic and gallic acids was observed. In another study, green tea was irradiated, resulting in no change in epicatechin and a 65% increase in catechin [31]. In contrast, UV irradiation resulted in a slight loss of chlorogenic acid and epicatechin in apple juice, but a slight rise in catechin was eventually observed [27]. However, the levels of these compounds did not change in the samples irradiated in this study.

### 3.4. Fatty Acid and Tocopherol Composition

Although fat is not the main component of peppercorns [32], they play an essential role in the organoleptic characteristics of food [33,34]. The main components of the fat fraction, the fatty acids, are sensitive to heat and other variables, which increase degradation reactions and can produce off-flavours [12,35].

The most abundant fatty acids found in all the samples were palmitic acid (C16:0), stearic acid (C18:0), oleic acid (C18:1), linoleic acid (C18:2), and alpha linolenic acid (C18:3) (Figure 4). Pink pepper was the sample that contained the highest levels of fatty acids (6400 mg/kg), followed by green (3800 mg/kg) and black pepper (2200 mg/kg). Notably, the sample with the lower amount of fatty acids (white pepper, 600 mg/kg) was the only one to result in a significant reduction of 22% of the total fatty acids after UV treatment. The major fatty acid in pink pepper was C18:2, whereas in black, green and white peppers it was C16:0. The second fatty acid in abundance was C18:1 in white and pink peppercorns, however in black and green pepper samples it was C18:2. Although no reports for green and pink pepper were found in this regard, the overall fatty acid profile of the black and white peppers matches previous results [36].

Concerning minor fatty acids, tetracosanoic acid (C24:0) was observed to be the highest in black, white and green pepper, followed by docosanoic acid (C22:0), tetradecanoic acid (C14:0), heptadecanoic acid (C17:0) and docosanoic acid (C20:0). Green pepper, contrary to black and white pepper, showed quantifiable levels of palmitoleic acid (C16:1) and 11-eicosenoic acid (C20:1). Again, pink pepper showed a different minor fatty acid profile with C14:0 being the most abundant.

Significant changes were observed after UV irradiation of peppercorns. In black pepper, only one major fatty acid was significantly affected by the UV treatment, decreasing by 13%. Some minor fatty acids were also reduced considerably, notably C24:0, which declined by 30%. This fatty acid was also observed to decrease similarly in both white and green peppers; however, many other fatty acids decreased by 8–34% in both types of peppers, although green peppers were generally less susceptible to UV light. In contrast, and despite pink pepper having almost 3 times more fat than *P. nigrum* peppercorns [37], it was not significantly affected by the UV treatment. Similarly, C18:3, a major omega-3 fatty acid, was not significantly altered in any of the peppers after the 90 min UV treatment.

It is well known that antioxidants present in foodstuffs delay the degradation of fatty acids [35], and the higher antioxidant capacity of pink pepper may have prevented their degradation [10,12]. Furthermore, the oxidative stability of oils has been observed to remain the same after UV irradiation due to the presence of tocopherols and polyphenols [38]. Other studies have investigated the effect of UV treatments on the composition of fatty acids [12,39,40]. However, none of these treatments was conducted with a reactor, and the agitation of peppercorns might have been partially or wholly responsible for the degradation of fatty acids.

The tocopherol composition was also observed to be affected by the treatment. As shown in Figure 5, alpha tocopherol, the predominant tocopherol in all three *P. nigrum* peppercorns, exhibited a decline of 20% and 26% in black and white pepper-treated samples, respectively. Nevertheless, a comparable decline in gamma tocopherol was exhibited by green pepper, which is consistent with the findings of another article that examined the degradation of tocopherols following UV irradiation [12]. The presence of delta tocopherol was exclusively observed in pink pepper, which showed no substantial alterations in any of the tocopherol types. As observed with other compounds, the UV light might have degraded alpha and gamma tocopherols, depending on the matrix, due to the cleavage of double bonds hit by UV photons, the formation and action of ROS and photoisomerization [2]. Vitamin E, also known as tocopherol, is essential for the proper functioning of the metabolism [41]. However, not all isomers are equally valuable, as alpha tocopherol has been observed to be far more effective in its antioxidant functions [42], and preserving this isomer is more critical.

### 3.5. Sensory Analysis and Relation with Chemical Changes

To evaluate the overall impact of UV-C treatment on peppercorn quality, chemical analyses were complemented with sensory tests. To date, no sensory analysis has been performed on pepper after UV treatment. Although compositional changes can indicate potential alterations in sensory perception, direct sensory evaluation is essential to determine whether these changes are perceptible to consumers.

Following the completion of the sensory tests, no significant differences (*n* = 19) were observed in any of the peppers or parameters after 90 min of UV treatment using the reactor (Table 3). Despite previous findings indicating substantial changes in total phenolic content and antioxidant activity [10], and the present study’s observations of alterations in colour parameters, volatile compounds, organic acids, and fatty acids, these chemical modifications did not translate into perceptible sensory differences. Collected feedback from participants regarding sensory attributes revealed contradictory observations among individuals who marked correct options, resulting in no significant differential characteristics.

Although a ΔE value of 1 is generally considered the threshold for perceiving a difference between two uniform colours, the complexity of food matrices can obscure such distinctions [43]. In fact, despite clearly surpassing this threshold, the participants in the sensory analysis could not observe colour differences. This result suggests that the inherent variability in the texture and pigmentation of peppercorns may limit the effectiveness of instrumental colour measurements in predicting human perception. This limitation is attributed to the fact that the quantitative data derived from such techniques may not accurately represent the visual perception of equivalent reference atlas colours [44]. As a result, the colour values might be very different for two similar multicoloured food products, and the visual complexity of multicoloured food products may require a greater ΔE for perceptible differences to emerge.

Regarding volatiles, small changes in the components may not have exceeded the detection threshold for complex odours, hindering their differentiation. According to Jagella & Grosch [45], the most potent odorants in black pepper are α-pinene (fresh, piney), myrcene (earthy, herbal), and linalool (floral, sweet), and it is believed that its smell is optimum when monoterpenes are high and pinene is low [46]. In this study, no significant changes were observed in these key volatiles following UV treatment, which likely explains the absence of perceptible differences in smell. Furthermore, 2,3-diethyl-5-methylpyrazine and 2-isopropyl-3-methoxypyrazine, which produce off-flavours [45], were not detected. In the case of white pepper, linalool and limonene (citric) have been observed to be intensely involved in the overall odour of white pepper, and p-cresol (medicinal, tar-like), indole (faecal) and skatole (faecal) are suggested to be responsible for the off-flavours [47]. Although no off-flavour components were detected and linalool did not undergo significant changes, limonene decreased to almost half of its original concentration, which should have altered the overall smell; however, it did not. Some of the less susceptible compounds, such as α-copaene, caryophyllene, β-bisabolene, linalool, β-elemene, and humulene, might have contributed to the characteristic odours in the peppers, resulting in no perceptible changes. The perception of food smell is a complex process, and many factors can coincide, such as odour blending, overshadowing, and synergistic interactions [48].

Flavour perception is similarly multifaceted. Organic acids play a key role in flavour, particularly in spices, where they can contribute to bitterness and pungency, as well as influence colour [49]. Polyphenols and organic acids were not as affected as volatile compounds, but the same result was obtained: no perceptible change in flavour was observed. Although the intrinsic difficulty of the test to evaluate the flavour of pepper hinders its capability of detecting differences [50], only two to three compounds were heavily influenced by the UV treatment in each pepper, which were minor compounds. Furthermore, piperine is one of the major flavour components of *P. nigrum* peppers, and UV treatments have been observed to result in little to no changes [7,10,51]. While fatty acids also contribute to flavour, their low concentration in peppercorns and the complexity of the matrix likely minimised their sensory impact, despite observed compositional changes.

### 3.6. Limitations, Industrial Implications, and Future Research Directions

This study provides a detailed assessment of the chemical and sensory effects of UV-C irradiation on various types of peppercorns using a scalable mechanical drum reactor. However, certain limitations should be acknowledged. The sensory analysis was conducted with consumer panellists, and despite the convenience of this method and its higher resemblance to that of general consumers, it may have limited the detection of subtle differences in smell or flavour. Additionally, although the study encompassed a broad range of chemical markers, it did not investigate the long-term stability of these compounds during storage, which could impact product shelf life and consumer perception over time.

From an industrial standpoint, the findings are highly promising. According to the obtained results, the use of up to 90 min of treatment or 30 J/cm^2^ using a rotary drum reactor could provide a significant disinfection of up to 2.6 log CFU/g, as observed in a previous work [10], without producing perceptible changes for the consumer, improving food safety with no significant repercussion in quality. The use of UV-C irradiation in a drum reactor offers a non-thermal, energy-efficient, and scalable solution for spice processing. The reactor design is compatible with existing production lines, requiring minimal adaptation, and the treatment does not compromise key sensory attributes such as colour, smell, or flavour. UV irradiation is particularly attractive for high-value spices where sensory integrity is critical to marketability.

Future research should focus on validating the process under continuous industrial conditions, assessing its integration into automated lines, and evaluating the economic feasibility at scale. Studies on other spice varieties, the combination of non-thermal treatments, the use of custom-wavelength LEDs, and the use of professional panellists would also enhance the reliability and applicability of this technology across the food industry.

## 4. Conclusions

This study evaluated the chemical and sensory impact of UV-C irradiation using a scalable mechanical drum reactor on four types of peppercorns: black, white, green, and pink. The results demonstrated that while UV-C treatment induced measurable changes in specific chemical parameters, such as volatile compounds, organic acids, fatty acids, and tocopherol content, these alterations were generally minor and varied depending on the type of pepper. Notably, based on the results, black and green peppercorns were less prone to suffer changes in their volatiles and phenolic profiles after the 90 min UV treatment. This effect is likely due to the protective role of the pericarp, whereas white and pink peppercorns were more susceptible.

Despite these chemical modifications, sensory analysis revealed no statistically significant differences in colour, smell, or flavour perception among treated and untreated samples, suggesting that UV-C irradiation using a drum reactor does not compromise the sensory quality of peppercorns in a manner that is perceptible to consumers.

The findings support the feasibility of UV-C irradiation as a non-thermal, scalable, and effective disinfection method for peppercorns, offering a promising alternative to conventional thermal treatments. This approach enhances food safety while preserving the sensory and nutritional quality of the product, making it suitable for industrial applications.

## Figures and Tables

**Figure 1 foods-14-03056-f001:**
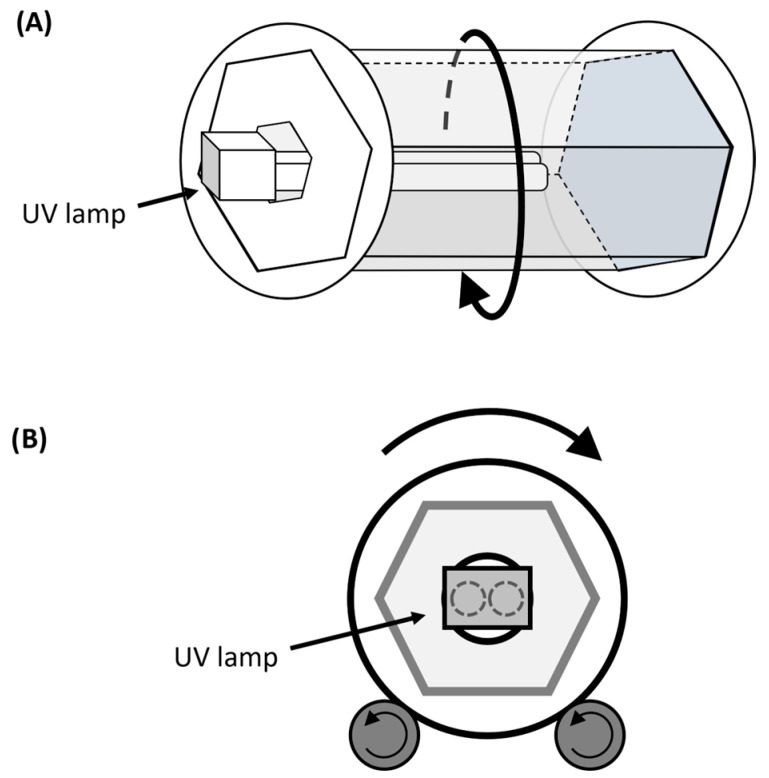
Schematic diagram of the UV reactor. Side-front view (**A**) and front view (**B**). Note: arrows indicate the direction of rotation of the reactor.

**Figure 2 foods-14-03056-f002:**
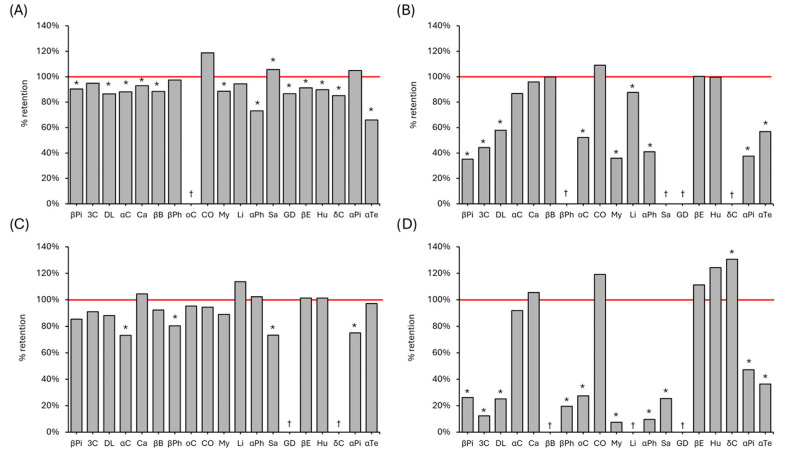
Retention of volatile compounds in black (**A**), white (**B**), green (**C**) and pink (**D**) peppercorns after the 90 min UV treatment relative to an untreated counterpart. Notes: *: Significant differences (*p* < 0.05) between UV-treated and untreated samples. †: Below the limit of detection. βPi: β-Pinene. 3C: 3-Carene. DL: D-Limonene. αC: α-Copaene. Ca: Caryophyllene. βB: β-Bisabolene. βPh: β-Phellandrene. oC: o-Cymene. CO: Caryophyllene oxide. My: Myrcene. Li: Linalool. αPh: α-Phellandrene. Sa: Sabinene. GD: Germacrene-D. βE: β-Elemene. Hu: Humulene. δC: δ-Cadiene. αPi: α-Pinene. αTe: α-Terpinolene.

**Figure 3 foods-14-03056-f003:**
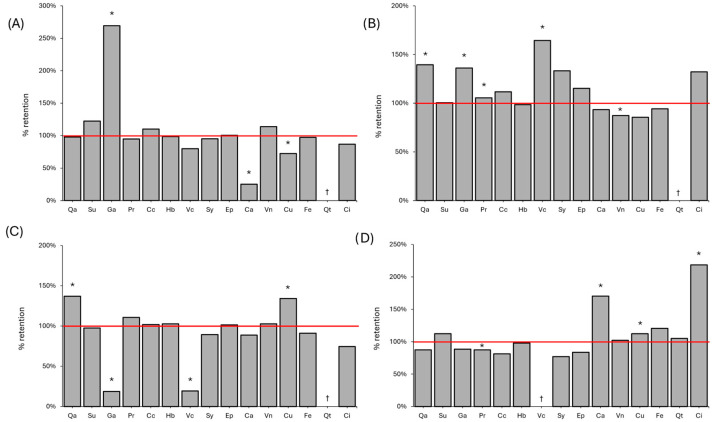
Retention of organic acids in black (**A**), white (**B**), green (**C**) and pink (**D**) peppercorns after the 90 min UV treatment relative to an untreated counterpart. Notes: *: Significant differences (*p* < 0.05) between UV-treated and untreated samples. †: Below the limit of detection. Qa: quinic acid. Su: succinic acid. Ga: gallic acid. Pr: protocatechuic acid. Cc: catechin. Hb: p-hydroxybenzoic acid. Vc: vanillic acid. Sy: syringic acid. Ep: epicatechin. Ca: caffeic acid. Vn: vanillin. Cu: coumaric acid. Fe: ferulic acid. Qt: quercetin. Ci: cinnamic acid.

**Figure 4 foods-14-03056-f004:**
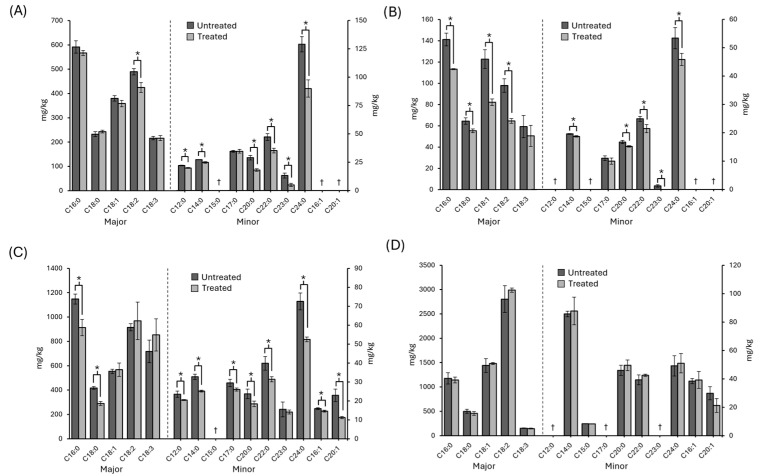
Untreated and UV-treated black (**A**), white (**B**), green (**C**) and pink (**D**) pepper fatty acid composition (mean ± SD). Notes: Two scales were used on the *Y* axis for major and minor fatty acids, divided by a dotted line. *: Significant differences (*p* < 0.05) between treated and untreated samples. †: Below the limit of detection. C12:0—dodecanoic acid. C14:0—tetradecanoic acid. C15:0—pentadecanoic acid. C16:0—palmitic acid. C17:0—heptadecanoic acid. C18:0—stearic acid. C20:0—eicosanoic acid. C22:0—docosanoic acid. C23:0—tricosanoic acid. C24:0—tetracosanoic acid. C16:1—palmitoleic acid. C18:1—oleic acid. C20:1—11-eicosenoic acid. C18:2—linoleic acid. C18:3—alpha linolenic acid.

**Figure 5 foods-14-03056-f005:**
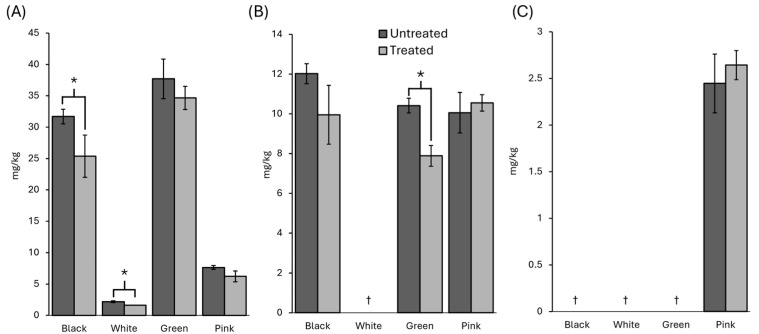
Alpha (**A**), gamma (**B**) and delta (**C**) tocopherol concentration in black, white, green and pink untreated and UV-treated pepper (mean ± SD). Notes: *: Significant differences (*p* < 0.05) between treated and untreated samples. †: Below the limit of detection.

**Table 1 foods-14-03056-t001:** Gradient and solvent composition of the UHPLC experiment.

Time	A	B
6 min	65%	35%
10 min	55%	45%
12 min	90%	10%

Notes: A: 99.9% water, 0.1% formic acid. B: 99.9% acetonitrile, 0.1% formic acid.

**Table 2 foods-14-03056-t002:** Average ± SD of CIELAB colour parameters (L*, a*, b*), total colour difference (∆E) and RGB visual representation of untreated and treated samples.

Type of Pepper	Colour Parameter	Control	Treated
Black	L*	33.1 ± 0.8 a	32.4 ± 0.4 a
	a*	2.9 ± 0.2 a	3.28 ± 0.05 b
	b*	9.0 ± 0.5 a	7.2 ± 0.2 b
	∆E		1.9 ± 0.3
	RGB		
Green	L*	35.2 ± 0.3 a	35.9 ± 0.4 b
	a*	0.1 ± 0.1 a	−0.1 ± 0.1 b
	b*	14.4 ± 0.5 a	13.6 ± 0.4 a
	∆E		1.2 ± 0.2
	RGB		
White	L*	40.2 ± 0.6 a	42.4 ± 0.4 a
	a*	2.3 ± 0.1 a	2.44 ± 0.03 b
	b*	12.1 ± 0.3 a	11.4 ± 0.2 a
	∆E		1.2 ± 0.1
	RGB		
Pink	L*	26.6 ± 0.6 a	25.4 ± 0.6 b
	a*	13.1 ± 0.5 a	10.5 ± 0.5 b
	b*	9.4 ± 0.7 a	6.0 ± 0.6 b
	∆E		4.4 ± 0.7
	RGB		

Notes: Different letters within the same row indicate significant differences (*p* < 0.05) between mean values.

**Table 3 foods-14-03056-t003:** Number and percentage of correct answers in the triangular test of colour, smell and flavour of untreated and UV-treated black, white, green and pink pepper.

Sensory Parameter	Black Pepper	White Pepper	Green Pepper	Pink Pepper
Colour	17 (43%)	17 (43%)	16 (40%)	13 (33%)
Smell	12 (30%)	18 (45%)	14 (35%)	14 (35%)
Flavour	10 (25%)	17 (43%)	9 (23%)	18 (45%)

Notes: Significant differences (*p* < 0.05) were established as the number of correct answers being 19 or more, according to the ISO 4120:2021 norm. Notes: Total sample size: *n* = 58. Number of valid responses per sample: *n* = 40.

## Data Availability

The raw data supporting the conclusions of this article will be made available by the authors on request.

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
