# Peer review of "Impact of a UV-C Scalable Reactor on the Chemical and Sensory Quality of Peppercorns"

_foods, 2025, doi:10.3390/foods14173056_

Round 1
Reviewer 1 Report
Comments and Suggestions for Authors
The authors should address the following issues:
Abstract
- The two sentences in lines 17-19 seem inconsistent. While the first sentence indicates that the overall sensorial alterations varied depending on the pepper types, the latter states that UV-C exposure did not result in any sensorial differences regardless of the pepper types.
- The final statement (lines 21-22) makes a generalized claim of UV-C irradiation for spice treatment, whereas this research investigated only peppercorn. Therefore, the conclusion appears to be overstated.
Materials and methods
- The English letters in the words “Quinic Acid” (line 71), “Epicatechin” (line 76), and “Catechin” (line 77) should be changed from uppercase to lowercase.
- Scientific name (line 88) should be italicized.
- For colour measurement, the “crushed” samples were used (line 112). Please clarify how particle size was standardized or controlled (e.g. crushing method, sieve mesh, particle size range). Additionally, were the samples used for sensory analysis also crushed? If so, were they matched the same particle-size distribution?
- The authors mix the Hunter (L, a, b) and CIELAB (L*, a*, b*) colour systems. Please clarify which system was used and ensure consistency in reporting those colour values.
- Since there are no other subsections under ‘Sensory Analysis’, the ‘Participants’ section should be integrated into ‘Sensory Analysis’ rather than presented as a separate subsection.
- The section on ‘Sensory Analysis’ should provide details on the following aspects:
- It should be clearly stated that all triangle tests were performed specifically to determine if differences existed between UV-C irradiated and non-irradiated peppercorn samples.
- Please indicate the total number of sample sets being tested in this research and the number of sample sets each participant was required to evaluate. In cases where the evaluation was conducted over multiple days, please specify the number of sample sets each participant was required to evaluate per day and the rest period between sample sets.
- The methods section states that there were 58 participants; however, the results indicate that only 40 panellists evaluated each sample set. Therefore, the number of panellists per sample set should be clearly specified in the methods section. Additionally, please clarify whether the panellists who evaluated each sample set for colour, smell and flavour were the same individuals.
- For flavour evaluation, please specify the rinse agent used for palate cleansing to minimize carry-over effects caused by the pungency and heat sensation of peppercorns. - Line 209, the term “control” should not be used in reference to a sample because triangle test does not involve a control sample; it consists only of an odd sample and two identical samples.
- Lines 217-218, the phrase “participant who presented a condition” is unclear. Please provide details about the specific condition being referred to.
- Line 224, the t-test used should be specified as either independent or dependent t-test.
- The method used to analyze significant differences between samples according to the ISO 4120:2021 should be briefly explained.
Results and Discussion
- For volatile composition, the key odour-active compounds of peppercorns should be mentioned. The discussion should explain the significance of each of these compounds and discuss how increases or decreases in their concentrations might affect aroma and flavor of peppercorns.
- The discussion in lines 291-299 does not present supporting experimental results and the corresponding data collection details are not given in the Methods section. Please include the relevant results and methodological information to support these statements.
- The manuscript reports changes in flavonoids, phenolic acids, organic acids and fatty acids, comparing before and after UV-C irradiation, but does not discuss their relevance to peppercorn quality. The discussion should explain the significance of these compounds and discuss how increases or decreases in their concentrations are expected to impact overall quality of peppercorns.
- The minimum number of correct responses required to declare significant differences between samples in triangle tests should be specified both in the text and as a footnote in Table 3.
- Line 423, the authors indicated that drawing of a solid conclusion is not possible due to contradictory observations among individuals. Please clarify this. In fact, the outcomes of triangle test are based on collective panellist responses, not on individual panellist response.
- Line 439, it is unclear how the color difference value of 4 was obtained.
- Line 474, the authors indicated that using untrained panellists was one of the limitations of this research because these panellists may not be able to detect subtle differences in aroma and flavor between samples. Therefore, it is unclear why the authors chose to use untrained panellists without sensory acuity screening in the first place. Please clarify the rational for this choice.
Tables and Figures
- The titles of all tables and figures should be revised to clearly reflect that they present quality parameters of peppercorns, comparing UV-C irradiated and non-irradiated samples.
- Table 1, use a period (.) instead of a comma (,) when presenting decimal values. In addition, the number of decimal places should be consistent for all colour values.
- Tables S1 and S2, the number of decimal places should be consistent for all values. The results should also include comparisons of means to indicate whether differences between samples are statistically significant.
Comments on the Quality of English Language
The quality of English language of the manuscript is acceptable but could be improved to more clearly express the research.
Author Response
Thank you very much for taking the time to review this manuscript. We believe your suggestions have significantly improved the quality of the manuscript. Please find detailed responses below and the corresponding corrections highlighted in red in the re-submitted files.
Comment 1: The two sentences in lines 17-19 seem inconsistent. While the first sentence indicates that the overall sensorial alterations varied depending on the pepper types, the latter states that UV-C exposure did not result in any sensorial differences regardless of the pepper types.
Response 1: Thank you for your observation. You are correct. It was expressed as if sensory qualities varied among pepper types. The sentences were rewritten.
Comment 2: The final statement (lines 21-22) makes a generalized claim of UV-C irradiation for spice treatment, whereas this research investigated only peppercorn. Therefore, the conclusion appears to be overstated.
Response 2: “Spices” was replaced by “pepper”.
Comment 3: The English letters in the words “Quinic Acid” (line 71), “Epicatechin” (line 76), and “Catechin” (line 77) should be changed from uppercase to lowercase. Scientific name (line 88) should be italicized.
Response 3: These changes were implemented.
Comment 4: For colour measurement, the “crushed” samples were used (line 112). Please clarify how particle size was standardized or controlled (e.g. crushing method, sieve mesh, particle size range). Additionally, were the samples used for sensory analysis also crushed? If so, were they matched with the same particle-size distribution?
Response 4: A clarification was included in lines 111-112 and 204.
Comment 5: The authors mix the Hunter (L, a, b) and CIELAB (L*, a*, b*) colour systems. Please clarify which system was used and ensure consistency in reporting those colour values.
Response 5: Thank you for the comment. The colour units were standardised according to CIELAB values.
Comment 6: Since there are no other subsections under ‘Sensory Analysis’, the ‘Participants’ section should be integrated into ‘Sensory Analysis’ rather than presented as a separate subsection.
Response 6: The suggestion was implemented.
Comment 7: The section on ‘Sensory Analysis’ should provide details on the following aspects:
- It should be clearly stated that all triangle tests were performed specifically to determine if differences existed between UV-C irradiated and non-irradiated peppercorn samples.
- Please indicate the total number of sample sets being tested in this research and the number of sample sets each participant was required to evaluate. In cases where the evaluation was conducted over multiple days, please specify the number of sample sets each participant was required to evaluate per day and the rest period between sample sets.
- The methods section states that there were 58 participants; however, the results indicate that only 40 panellists evaluated each sample set. Therefore, the number of panellists per sample set should be clearly specified in the methods section. Additionally, please clarify whether the panellists who evaluated each sample set for colour, smell and flavour were the same individuals.
- For flavour evaluation, please specify the rinse agent used for palate cleansing to minimize carry-over effects caused by the pungency and heat sensation of peppercorns.
Response 7: Thank you very much for your suggestions that improve the quality of the methodology. All of them were clarified in section 2.8.
Comment 8: Line 209, the term “control” should not be used about a sample because triangle test does not involve a control sample; it consists only of an odd sample and two identical samples.
Lines 217-218, the phrase “participant who presented a condition” is unclear. Please provide details about the specific condition being referred to.
Line 224, the t-test used should be specified as either independent or dependent t-test.
The method used to analyze significant differences between samples according to the ISO 4120:2021 should be briefly explained.
Response 8: All the suggestions were implemented and clarifications were made in the text and highlighted in red.
Comment 9: For volatile composition, the key odour-active compounds of peppercorns should be mentioned. The discussion should explain the significance of each of these compounds and discuss how increases or decreases in their concentrations might affect the aroma and flavour of peppercorns.
Response 9: All the analysed volatile compounds (nineteen) in this study have been reported in previous studies as the main components, and a few of them as key odorants for pepper. This is discussed later in lines 453-461. However, the characteristics of these odorant compounds were added.
Comment 10: The discussion in lines 291-299 does not present supporting experimental results, and the corresponding data collection details are not given in the Methods section. Please include the relevant results and methodological information to support these statements.
Response 10: The methodology of the agitated control was moved to the methodology section (2.5) and rewritten for the sake of clarity. More details were included in the results, and numerical results are now also shown in Table S1.
Comment 11: The manuscript reports changes in flavonoids, phenolic acids, organic acids and fatty acids, comparing before and after UV-C irradiation, but does not discuss their relevance to peppercorn quality. The discussion should explain the significance of these compounds and discuss how increases or decreases in their concentrations are expected to impact overall quality of peppercorns.
Response 11: The impact of UV light on the quality of pepper (flavonoids, phenolic acids, organic acids and fatty acids) is discussed in the sensory analysis section, as we thought it would be clearer to discuss the results of the sensory analysis together with changes in the chemical components and their relationship with sensory attributes. Although it is not the only way to discuss the results, we believe this approach is preferable because it enhances our understanding of the treatment's impact on sensory quality.
Comment 12: The minimum number of correct responses required to declare significant differences between samples in triangle tests should be specified both in the text and as a footnote in Table 3.
Response 12: The suggestion was implemented in the methodology and also in the discussion.
Comment 13: Line 423, the authors indicated that drawing of a solid conclusion is not possible due to contradictory observations among individuals. Please clarify this. In fact, the outcomes of triangle test are based on collective panellist responses, not on individual panellist response.
Response 13: The qualitative feedback reported by the participants was either neutral or similar to one direction (e.g.: more smell) or the other (e.g. less smell). A brief clarification was added.
Comment 14: Line 439, it is unclear how the color difference value of 4 was obtained.
Response 14: It was based on empirical observations. However, it has been removed.
Comment 15: Line 474, the authors indicated that using untrained panellists was one of the limitations of this research because these panellists may not be able to detect subtle differences in aroma and flavor between samples. Therefore, it is unclear why the authors chose to use untrained panellists without sensory acuity screening in the first place. Please clarify the rationale for this choice.
Response 15: A brief clarification was added in the Limitations section.
Comment 16: The titles of all tables and figures should be revised to clearly reflect that they present quality parameters of peppercorns, comparing UV-C irradiated and non-irradiated samples. Table 1, use a period (.) instead of a comma (,) when presenting decimal values. In addition, the number of decimal places should be consistent for all colour values.
The titles and content of the tables and figures were adapted according to the suggestion.
Comment 17: Tables S1 and S2, the number of decimal places should be consistent for all values. The results should also include comparisons of means to indicate whether differences between samples are statistically significant.
Response 17: The decimals in tables S1 and S2 were set according to the significant figures’ rules, depending on the first important figure of the SD. The statistically significance of the supplementary data was added.
Comment 18: The quality of English language of the manuscript is acceptable but could be improved to more clearly express the research.
Response 18: A general revision and improvement of the language was conducted.
Reviewer 2 Report
Comments and Suggestions for Authors
The Manuscript ID: foods-3824577 titled “Impact of a UV-C scalable reactor on the chemical and sensory quality of peppercorns” is an interesting article where the UV-C radiation is used inside a rotary drum to measure the effect on chemical and sensory properties of UV-C on black, white, green and pink peppercorns. However, the manuscript must be improved, the authors should consider the following:
Line 92 does not mention the material of the hexagonal rotary drum, please provide this information.
In methodology for UV-C treatment please insert a paragraph explaining how the dose 29.8 103 ± 0.5 J/cm2 was obtained.
In methodology for UV-C treatment please include the load (mass of material used) for each experiment
Line 103 and 104 cm2 should be cm2
Line 112 samples were crushed, please mention the particle size used.
Lines 146 and 171. Provide X g instead of rpm
Line 190 Eliminate fluorescence irradiation, substitute it for absorption.
In table 2 the use of coma (,) and (.) is confusing. Use through all document (.) or (,) but it must be the same.
Table 2. The value of parameter “a” for green p. is the same as SD. Please check this data
In figure 2 D check the recovery, specifically for δC: δ-Cadiene, it is too high. Approximately 130%, this value in terms of quality control is unacceptable
In volatile composition include the possible mechanisms or reactions that occur provoking the increasing or decreasing in the concentration (Retention) of these compounds
For organic acids, how is possible % R > 120 (e.g. 275% for gallic acid in white p). Please check your results. If the results are the same, explain the possible formation mechanisms of them. mentions the possible mechanisms of formations of specific organic acids. Also include information about the mechanisms involved in the decreasing of some organic acids.
Line 353 for each abbreviation is necessary to provide first the meaning, e.g. palmitic acid (C16:0)
For fatty acids and tocopherols is necessary a deeply discussion about the results, explain possible reactions for formation or degradation of some compounds
Author Response
Thank you very much for taking the time to review this manuscript. Please find detailed responses below and the corresponding corrections highlighted in red in the re-submitted files.
Comment 1: Line 92 does not mention the material of the hexagonal rotary drum, please provide this information.
Response 1: Thank you for the suggestion. The material was specified.
Comment 2: In methodology for UV-C treatment please insert a paragraph explaining how the dose 29.8 103 ± 0.5 J/cm2 was obtained.
Response 2: The dose was obtained under the same conditions as the samples were treated, using a radiometer sensor, measuring the average irradiance. Since irradiance is measured in mW/cm2, you can calculate the dose (J/cm2) by multiplying the time. It is further explained in lines 98-105.
Comment 3: In methodology for UV-C treatment please include the load (mass of material used) for each experiment.
Response 3: The load is now included in the methodology.
Comment 4: Line 103 and 104 cm2 should be cm2.
Response 4: The change was implemented throughout the manuscript.
Comment 5: Line 112 samples were crushed, please mention the particle size used.
Response 5: More specifications were provided in lines 112-113.
Comment 6: Lines 146 and 171. Provide X g instead of rpm
Response 6: The suggestion was implemented in line 146. However, in line 171 RPM are used to report the speed of a magnetic stirrer. We acknowledge the lack of clarity there and specified it.
Comment 7: Line 190 eliminate fluorescence irradiation, substitute it for absorption.
Response 7: The suggested change was implemented.
Comment 8: In table 2 the use of coma (,) and (.) is confusing. Use through all document (.) or (,) but it must be the same.
Response 8: The suggested change was implemented throughout the manuscript.
Comment 9: Table 2. The value of parameter “a” for green p. is the same as SD. Please check this data.
Response 9: Thank you for the meticulous review of the manuscript. In this case, both a* and b* are parameters that go from -127 to 127, so 0.1 ± 0.1 is an acceptable outcome. We also revised the data and checked that everything was right.
Comment 10: In figure 2 D check the recovery, specifically for δC: δ-Cadiene, it is too high. Approximately 130%, this value in terms of quality control is unacceptable
Response 10: Maybe there was a misunderstanding. The graph does not show recovery, but retention. The formula employed was: [concentration of cadiene in the UV-treated sample/concentration of cadiene in the untreated sample] x 100. This provides a relative unit (instead of very different intensity numbers that could not be combined in one graph) to assess the changes resulting from the UV treatment easily. A clarification was added in the methodology to avoid confusion. As observed in other compounds in many other vegetables, UV light can promote the formation of compounds due to the degradation of others or the synthesis due to light stress. UV-induced hormesis has been widely reported, and it depends on the compound and the matrix. Our previous research confirms that finding. Here is an unrelated example of this phenomenon: Bravo, S., García-Alonso, J., Martín-Pozuelo, G., Gómez, V., Santaella, M., NavarroGonzález, I., et al. (2012). The influence of UV-C hormesis on lycopene, β-carotene, and phenolic content and antioxidant activity of breaker tomatoes. Food Research International, 49,296–302.
Comment 11: In volatile composition include the possible mechanisms or reactions that occur provoking the increasing or decreasing in the concentration (Retention) of these compounds
Response 11: An explanation was provided in lines 299-302.
Comment 12: For organic acids, how is it possible % R > 120 (e.g. 275% for gallic acid in white p). Please check your results. If the results are the same, explain the possible formation mechanisms of them. mentions the possible mechanisms of formations of specific organic acids. Also include information about the mechanisms involved in the decreasing of some organic acids.
Response 12: As explained in response 10, that is a possible outcome. An explanation was added in lines 359-364.
Comment 13: Line 353 for each abbreviation is necessary to provide first the meaning, e.g. palmitic acid (C16:0)
Response 13: The suggestion was implemented.
Comment 14: For fatty acids and tocopherols is necessary a deeply discussion about the results, explain possible reactions for formation or degradation of some compounds
Response 14: A explanation was added to the discussion.
Reviewer 3 Report
Comments and Suggestions for Authors
Please find the attached file

Author Response
Thank you very much for taking the time to review this manuscript. Please find detailed responses below and the corresponding corrections highlighted in red in the re-submitted files.
Comment 1: Add some result numbers to the abstract.
Response 1: Some numbers were added to the abstract, but the word limitation in the abstract prevents us from adding more.
Comment 2: "Keywords" Please avoid the same terms used in the title as much as possible.
Response 2: Thank you for the suggestion. Keywords were modified according to it.
Comment 3: What is the harvest period or purchase of the raw material (month, year)?
Response 3: The date of the collection of the peppers is given in line 86 (year 2023). No more information than that was provided by the manufacturers.
Comment 4: What is the name of the pepper, variety on which the experiments were conducted? (Varieties must be identified and written scientifically (in italics)).
Response 4: The scientific names of the species were italicised throughout the manuscript. We know the country of origin, species and type of pepper, but the manufacturer gave no variety name.
Comment 5: Lines 103, and 506: W/cm2 must be changed to W/cm2- Lines 104, 242, and 279 J/cm2 must be changed to J/cm2
Response 5: The suggestion was implemented.
Comment 6: A definition must be added for L1, L2, a1, a2, b1, and b2
Response 6: The suggestion was implemented.
Comment 7: Lines 117: Why is there no star on the symbol (L)? Symbols must be placed with or without a star throughout the manuscript, including the equation. Lines 246: Why is there no star on the symbol (L)? Symbols must be placed with or without a star throughout the manuscript, and Table 2
Response 7: The use of the star was standardised.
Comment 8: Lines 133, 134, and 157: 70 ºC must be changed to 70 oC
Response 8: The suggestion was implemented.
Reviewer 4 Report
Comments and Suggestions for Authors
The study is quite relevant for the overall food safety of species worldwide trade; mostly if it would be applied to an industrial scale.
However in terms of the sensory analysis it had a lot of limitations which will been exposed in an attached file in order to be clarified and/or corrected to allow the publish of the article

Author Response
Thank you very much for taking the time to review this manuscript. Your thorough revision and comments were beneficial in improving the quality of the manuscript. Please find detailed responses below and the corresponding corrections highlighted in red in the re-submitted files.
Comment 1: L17 - It should be “sensory” instead of “sensorial”
Response 1: The wording was changed.
Comment 2: L 65 - “smell and aroma” or just “smell”, since in the abstract was mentioned aroma in L 18
Response 2: The wording was changed in the abstract.
Comment 3: L87 + L88 - Species names in italics, and review in ALL article
Response 3: The suggestion was implemented.
Comment 4: L115 + L116 - Which calibration values are??? In what colour scale?? Hunter scale??
Response 4: The calibration values of the blank were added to the methodology. The specification of the use of CIELAB scale was added in line 118.
Comment 5: L197 + L198 - “ (...) three sensory attributes: colour, smell, and flavour.” - It has to be clear which of the sensory attributes is going to be analyzed in this study: if SMELL or if AROMA, because they are not the same!
Response 5: Aroma was replaced by smell.
Comment 6: L 198 + L199 - How many triangulars were presented to each panelist with 4 samples to taste?! This is not clear in M&M. L201 + L202+L203 - Were presented 3 vials containing 1g/peppercorns for the smell, but there were 4 different peppers as samples; confuse...CLARIFY please.
Response 6: More details were added in the sensory analysis section to clarify the methodology.
Comment 7: L 203 - Rice for the flavour? Can you indicate the reference ?
Response 7: A reference was added in line 219
Comment 8: L 204 - What is warm?! How do you kept it warm?! At that range of TºC? In sensory
analysis all temperatures have to have at least a range...CLARIFY pleas
Response 8: More details were added in the section.
Comment 9: L217 to L219 - How was this selection done? Previous survey?!
Response 9: More details were added in this regard.
Comment 10: In resume: All sensory procedures is not very clear, mostly the results are not in agreement with the M&M presentation of the proposed techniques applied within theTriangular Method for the 3 attributes to characterize; and even he fact that all 4 samples were always analyized with a Control Sample with no UV treatment, and in sensory proccedures it was not clear how were presented all triangular sessions (each with JUST3SAMPLES) in order for all panellists to have the chance to taste 4 UV samples and 4 samples not UV treated.
Response 10: The sensory analysis section was completely rewritten to add details and make it more straightforward.
Comment 11: L 237 + L238 - Why do you comment the differences between the peppers if you had not analysed its statistically (in columns) – table 2 – L246; it was only done the statistical analysis in each pepper and between the control and the irradiated one (in rows)
Response 11: What we mean by that sentence is that the differences (e.g. an increase in a*) are the same across samples. Changes vary in significance and direction in the different samples. The changes are not the same in all samples. Some present significant differences in L*, others have a substantial increase in a* and others have a significant decrease in a*. Changes are not consistent between pepper types.
Comment 12: L 246 - NOT “colour change” BUT “Total Colour Difference” as mentioned properlly in M&M - L117
Response 12: The suggestion was implemented.
Comment 13: L 417 - As mentioned in L195 in the subchapter of Sensory analysis in M&M, the results are lack of being bonded to the M&M, since table 3 - L429, has very few explanations for n=40. And also, no results for the Control Sample, thus how it is possible to mention that there were no Significant differences between the treated a not treated samples for each peppercorns???
Response 13: A triangular test is a test where three samples or subsamples (one control and two treated samples or two treated samples and one control) are given to each participant to evaluate one of the attributes. It is not used to compare between pepper types, but to compare between two similar foodstuffs (e.g. untreated and treated black pepper). Therefore, every triangular test always contains control and UV-treated samples, for which the participants need to discern between the two. Also, not every participant performed the triangular tests for all four peppers; that is why the total number of participants is higher than the number of valid answers (n=40). However, we acknowledge the lack of clarity and have improved the sensory analysis methodology.
Comment 14: L 426 - sensory analysis and NOT sensorial analysis
Response 14: The suggestion was implemented.
Comment 15: In resume, concerning sensory analysis results discussion, another important data is the fact that in the universe of 58 candidates for sensory only 69% had correct responses to the Triangular sessions; it would be important to know how were designed the Triangular sessions in order to better understand the incorrect responses of the 31% of panellists.
Response 15: The first number in each cell of the Table 3 stands for the percentage of correct answers (45 to 23%), and the number in brackets stands for the number of correct answers (18 to 9), being the total of valid answers in all peppers n=40, as indicated in the first row. Although we believe the percentage of correct answers is sufficient to convey the sensory analysis results, the ISO 4120:2021 norm also requires expressing the results in total correct answers to assess significant differences. In this case, 19 correct answers were necessary to observe significant differences in any of the samples and sensory tests, so no significant differences were observed. However, we changed the format of the table to make it easier to understand.
Comment 16: L 469 - This is not an expression proper for handwriting; is more for the speech; should be replaced
Response 16: A general revision and improvement of the language was performed.
Comment 17: L505 + L506 - This detailed appreciation of sensory results related with a specific irradiation input should not be in the conclusions, fist because it was already stated in the discussion (where it belongs) and also due to the sensory limitations already mentioned previously.
Response 17: The conclusions were slightly modified according to the suggestion.
Round 2
Reviewer 2 Report
Comments and Suggestions for Authors The authors considered all observations and suggestions made with the aim of improving the manuscript